# Environmental Persistence of *Staphylococcus capitis* NRCS-A in Neonatal Intensive Care Units: Role of Biofilm Formation, Desiccation, and Disinfectant Tolerance

Marie Chavignon,[a] Ludivine Coignet,[a] Mélanie Bonhomme,[a] Marine Bergot,[a*] Anne Tristan,[a,b] Paul Verhoeven,[c,d] Jérôme Josse,[a] Frédéric Laurent,[a,b] Marine Butin[a,e]

[a]Centre International de Recherche en Infectiologie (CIRI), Team Pathogénie des Staphylocoques, INSERM U1111, Université Claude Bernard Lyon 1, CNRS, UMR5308, ENS Lyon, Lyon, France
[b]Institut des Agents Infectieux, Centre National de Référence des Staphylocoques, Hospices Civils de Lyon, Lyon, France
[c]Centre International de Recherche en Infectiologie (CIRI), Team GIMAP (Groupe sur l'Immunité des Muqueuses et Agents Pathogènes), INSERM U1111, Université Claude Bernard Lyon 1, CNRS, UMR5308, ENS Lyon, Lyon, France
[d]Department of Infectious Agents and Hygiene, University-Hospital of Saint-Etienne, Saint-Etienne, France
[e]Service de Néonatologie et Réanimation Néonatale, Hôpital Femme Mère Enfant, Hospices Civils de Lyon, Bron, France

**ABSTRACT** The clone *Staphylococcus capitis* NRCS-A is responsible for late-onset sepsis in neonatal intensive care units (NICUs) worldwide. Over time, this clone has evolved into three subgroups that are increasingly adapted to the NICU environment. This study aimed to decipher the mechanisms involved in NRCS-A persistence in NICUs. Twenty-six *S. capitis* strains belonging to each of the three NRCS-A clone subgroups and two other non-NRCS-A groups from neonates (alpha clone) or from adult patients ("other strains") were compared based on growth kinetics and ability to form biofilm as well as tolerance to desiccation and to different disinfectants. *S. capitis* biofilm formation was enhanced in rich medium and decreased under conditions of nutrient stress for all strains. However, under conditions of nutrient stress, NRCS-A strains presented an enhanced ability to adhere and form a thin biofilm containing more viable and culturable bacteria (mean 5.7 $\log_{10}$ CFU) than the strains from alpha clone (mean, 1.1 $\log_{10}$ CFU) and the "other strains" (mean, 4.2 $\log_{10}$ CFU) ($P < 0.0001$). The biofilm is composed of bacterial aggregates with a matrix mainly composed of polysaccharides. The NRCS-A clone also showed better persistence after a 48-h desiccation. However, disinfectant tolerance was not enhanced in the NRCS-A clone in comparison with that of strains from adult patients. In conclusion, the ability to form biofilm under nutrient stress and to survive desiccation are two major advantages for clone NRCS-A that could explain its ability to persist and settle in the specific environment of NICU settings.

**IMPORTANCE** Neonatal intensive care units (NICUs) host extremely fragile newborns, including preterm neonates. These patients are very susceptible to nosocomial infections, with coagulase-negative staphylococci being the species most frequently involved. In particular, a *Staphylococcus capitis* clone named NRCS-A has emerged worldwide specifically in NICUs and is responsible for severe nosocomial sepsis in preterm neonates. This clone is specifically adapted to the NICU environment and is able to colonize and maintain on NICU surfaces. The present work explored the mechanisms involved in the persistence of the NRCS-A clone in the NICU environment despite strict hygiene measures. The ability to produce biofilm under nutritional stress and to resist desiccation appear to be the two main advantages of NRCS-A in comparison with other strains. These findings are pivotal to provide clues for subsequent development of targeted methods to combat NRCS-A and to stop its dissemination.

**KEYWORDS** *Staphylococcus capitis* NRCS-A, neonatal intensive care units, environmental persistence, biofilm, disinfection

Address correspondence to Marine Butin, marine.butin@chu-lyon.fr.

*Present address: Marine Bergot, Equipe GAD, INSERM 1231, Université de Bourgogne, Dijon, France.

The authors declare no conflict of interest.

Neonatal intensive care units (NICUs) are specific settings where there are hospitalized neonates requiring medical care after birth, notably preterm neonates. Due to their delicate skin barrier and immune immaturity, preterm neonates are particularly vulnerable to bacterial infections (1). These infections occur just after birth due to the transmission of bacteria from the mother during delivery or involving bacteria from the environment of the NICU after a few days of life (late-onset sepsis) (2). These late-onset sepsis cases are frequently nosocomial and associated with mortality and neurosensorial and respiratory sequelae (3, 4). Among the most prevalent bacteria responsible for late-onset sepsis is the multidrug-resistant clone *Staphylococcus capitis* NRCS-A, which is remarkable due to its endemic diffusion in NICUs worldwide (5–7). This clone is divided into three subgroups (proto-outbreak I, proto-outbreak II, and outbreak) characterized by their chronological appearance/expansion and progressive adaptation for the NICU environment, notably due to the acquisition of mechanisms of resistance to antibiotics commonly used in NICUs such as methicillin and vancomycin (7, 8). Previous studies showed that the NRCS-A clone is able to colonize inert surfaces such as neonatal incubators and to persist in the NICU environment despite the use of chemical disinfection procedures, which fosters environmental dissemination and interpatient transmission (9–11).

Bacterial persistence on hospital surfaces has already been reported for other pathogens, and several mechanisms favoring such persistence have been identified. In particular, the ability to adhere to environmental surfaces by producing biofilm confers protection against environmental stresses, such as disinfectant molecules or antibiotics, and the host immune system (12, 13). In staphylococci, the biofilm matrix can be composed of polysaccharides associated with the production of polysaccharide intercellular adhesin (PIA) mediated by the *icaADBC* operon, bacterial secreted proteins, or extracellular DNA (eDNA) or teichoic acids, proportions of each differing from one strain to another (13). Coagulase-negative staphylococci collected from hospital surfaces have been shown to form more biofilm than strains isolated from the community environment, and it has been demonstrated that this was in part related to the expression of the *icaAD* and *bap* (involved in initial attachment) genes (14). Another mechanism of bacterial persistence on hospital surfaces despite standard hygiene measures is a decreased susceptibility to disinfectants, which can be related to changes in cell permeability, increased efflux pump expression, enzymatic degradation, mutations, or acquisition of resistance determinants by gene transfer (15). In that regard, Lepainteur et al. have demonstrated that 21 of 51 coagulase-negative staphylococcal isolates from late-onset sepsis in a NICU setting exhibited reduced susceptibility against at least one disinfectant, such as benzalkonium chloride, chlorhexidine, and acriflavine, and carried efflux pumps genes such as *qacA* and *qacB* (16). In addition, bacteria must also deal with desiccation in the hospital environment. Water being essential for biochemical reactions, desiccation negatively affects all biological functions, notably by leading to protein aggregation, architecture modifications, and/or enzymatic inactivation (17). Thus, *Staphylococcus aureus* was notably shown to be able to survive on dry plastic surfaces for more than 3 years (18). Such huge bacterial tolerance to desiccation involves several synergistic mechanisms, including biofilm formation (due to the high hydration of the matrix extracellular polysaccharides), import or synthesis of specific osmolytes such as trehalose (which contributes to maintain protein structure), and synthesis of heat shock proteins (17).

To date, it remains unclear how the NRCS-A clone is able to spread and become established in NICU environments worldwide. The aim of this study was to decipher the mechanisms involved in NRCS-A persistence inside NICU environments by investigating growth kinetics, biofilm production/features, and tolerance to desiccation and disinfectants based on a collection of clinical isolates belonging to the *S. capitis* species.

## RESULTS

**Kinetic growth.** The strains used in this study are described in Table 1, and their growth kinetics were evaluated in tryptic soy broth (TSB). In this medium, the median doubling times were 45.8 min for NRSC-A outbreak (mean, 47.1 $\pm$ 5.0 min), 43.8 min for NRSC-A proto-outbreak II (mean, 42.1 $\pm$ 3.6), 46.5 min for NRSC-A proto-outbreak I (mean, 47.7 $\pm$ 5.5), 49.8 min

**TABLE 1** Characteristics of the strains used in this study

| Strain | Patient | Yr of isolation | Country of isolation | Genes related to biofilm production | | | | | | | | | | Genes related to disinfectant tolerance | | | |
|---|---|---|---|---|---|---|---|---|---|---|---|---|---|---|---|---|---|
| | | | | bap | gehD | sarA | sdrH | sigB | icaADBC | icaR | alt | ebpS | ebh | qacA | qacB | qacJ | smr |
| *S. capitis* | | | | | | | | | | | | | | | | | |
| NRCS-A outbreak | | | | | | | | | | | | | | | | | |
| AD69[a] | Newborn | 2009 | France | – | + | + | + | + | + | + | + | + | + | – | – | – | – |
| AV74[a] | Newborn | 2009 | Brazil | + | + | + | + | + | + | + | + | + | + | – | + | – | – |
| AW19[a] | Newborn | 1999 | Norway | + | + | + | + | + | + | + | + | + | + | – | – | – | – |
| AW77[a] | Newborn | 2013 | New Zealand | – | + | + | + | + | + | + | + | + | + | + | – | – | – |
| BA22[a] | Newborn | 2014 | USA | – | + | + | + | + | + | + | + | + | + | – | – | – | – |
| BC08[a] | Newborn | 2015 | France | – | + | + | + | + | + | + | + | + | + | + | – | – | – |
| NRCS-A proto-outbreak II | | | | | | | | | | | | | | | | | |
| AW20 | Newborn | 1998 | The Netherlands | + | + | + | + | + | + | + | + | + | + | – | – | – | – |
| BC76[a] | Newborn | 2010 | South Korea | + | + | + | + | + | + | + | + | + | + | – | – | + | + |
| BD01 | Newborn | 2010 | South Korea | + | + | + | + | + | + | + | + | + | + | – | – | – | – |
| BD06 | Newborn | 2014 | South Korea | – | + | + | + | + | + | + | + | + | + | + | – | – | – |
| BI77[a] | Newborn | 2015 | USA | – | + | + | + | + | + | + | + | + | + | + | – | – | + |
| NRCS-A proto-outbreak I | | | | | | | | | | | | | | | | | |
| AK81[a] | Adult | 2012 | Australia | – | + | + | + | + | + | + | + | + | + | – | + | – | + |
| BD62 | Newborn | 2013 | Czech Republic | – | + | + | + | + | + | + | + | + | + | + | – | + | – |
| BG77 | Adult | 2014 | Finland | – | + | + | + | + | + | + | + | + | + | – | – | – | – |
| AL04 | Adult | 2012 | Australia | – | + | + | + | + | + | + | + | + | + | – | – | – | – |
| BA10[a] | Newborn | 2014 | Germany | – | + | + | + | + | + | + | + | + | + | + | – | – | + |
| *S. capitis* alpha clone | | | | | | | | | | | | | | | | | |
| AY57[a] | Newborn | 2014 | France | – | + | + | + | + | + | + | + | + | + | + | – | – | + |
| BD09 | Newborn | 2014 | France | – | + | + | + | + | + | + | + | + | + | + | – | – | – |
| AX72 | Newborn | 2014 | France | – | + | + | + | + | + | + | + | + | + | + | – | – | – |
| AY24 | Adult | 2013 | Greece | – | + | + | + | + | + | + | + | + | + | + | – | + | – |
| BD08[a] | Newborn | 2014 | France | – | + | + | + | + | + | + | + | + | + | + | – | – | – |
| Other *S. capitis* ("other strains") | | | | | | | | | | | | | | | | | |
| AW14[a] | Newborn | 1994 | Norway | – | + | + | + | + | + | + | + | + | + | – | – | – | – |
| BC70 | Adult | 2009 | South Korea | + | + | + | + | + | + | + | + | + | + | – | – | + | – |
| BD28 | Adult | 2014 | Taiwan | + | + | + | + | + | + | + | + | + | + | – | + | – | + |
| AY43 | Adult | 2013 | Belgium | – | + | + | + | + | + | + | + | + | + | – | – | – | + |
| BD18[a] | Adult | 2015 | France | – | + | + | + | + | + | + | + | + | + | + | – | – | + |
| *S. aureus* | | | | | | | | | | | | | | | | | |
| SH1000 | Control for biofilm formation | | | | | | | | | | | | | | | | |
| ATCC 25923 | Control for evaluation of disinfectant MICs and MBCs | | | | | | | | | | | | | | | | |

[a] Strains studied for the evaluation of the composition of the biofilm matrix in RPMI medium.

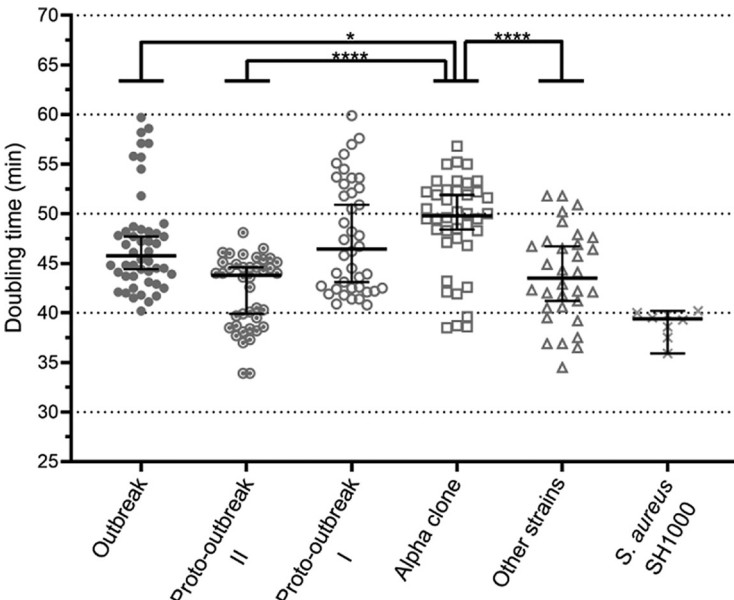

**FIG 1** Doubling times of *S. capitis* strains from the different groups in standard medium. The doubling time of each strain from the five *S. capitis* groups was calculated from growth kinetics in TSB medium. Results are shown as the median doubling time for each group with 95% confidence intervals. Statistical analyses were carried out using the Mann-Whitney nonparametric test with an $\alpha$ of 0.05.

for the alpha clone (mean, 49.0 $\pm$ 4.8 min), and 43.5 min for the "other strains" (mean, 43.7 $\pm$ 4.7 min) (Fig. 1). The median doubling time of the alpha clone was significantly higher than those of the NRSC-A outbreak ($P = 0.0130$), NRSC-A proto-outbreak II ($P < 0.0001$), and "other strains" ($P < 0.0001$).

**Biofilm formation.** An important heterogeneity in the level of biofilm formation was observed inside each *S. capitis* group, except for the alpha clone. This wide variation inside each group was particularly noticeable when the total biomass production was evaluated in glucose-supplemented TSB and under the hyperosmotic condition (NaCl-supplemented TSB) (Fig. 2A). Glucose supplementation and the hyperosmotic condition enhanced the global biofilm biomass, whereas the bacterial count remained stable (Fig. 2B). Under the condition of nutritional stress (RPMI medium), a decrease in biofilm production was observed for the five *S. capitis* groups, but the bacterial count in the biofilm of the NRSC-A clone (including the three subgroups, outbreak, proto-outbreak II, and proto-outbreak I) was significantly higher (mean, 5.7 $\log_{10}$ CFU) than in the groups of the alpha clone (mean, 1.1 $\log_{10}$ CFU) and the "other strains" (mean, 4.2 $\log_{10}$ CFU) ($P < 0.0001$).

**Composition of the biofilm matrix under nutritional stress.** The biofilms of *S. capitis* strains from the five groups were observed using confocal microscopy after staining of the different components of the biofilm matrix. These observations revealed a flat biofilm architecture with scattered small aggregates among adhered bacteria (Fig. 3). A wide heterogeneity in appearance was also noticed concerning the biofilm of strains from the same group. No difference in qualitative composition of biofilm matrix between the five groups of *S. capitis* was observed. Total DNA, eDNA, proteins, and polysaccharides were identified in the biofilm of each *S. capitis* group and generally colocalized. The abundance of polysaccharides was particularly high in bacterial aggregates and around isolated bacteria.

**Desiccation tolerance.** Intragroup variations between strains were observed particularly inside outbreak, proto-outbreak II, and the "other strains" groups at both 24 h and 48 h (Fig. 4). In these groups, one to two strains showed a visibly higher level of persistence than the other strains of their group (AD69 and AW77 for the outbreak group, BI77 for the proto-outbreak II group, and AW14 for the "other strains" group). Using the median percentage of persistence of each group, the alpha clone showed a significantly lower persistence, with a median persistence of 0.08%, than the four other groups at 24 h: outbreak, 0.44% ($P < 0.0001$); proto-outbreak II, 0.45% ($P < 0.0001$); proto-outbreak I, 0.29% ($P = 0.0007$); and "other strains," 0.35%

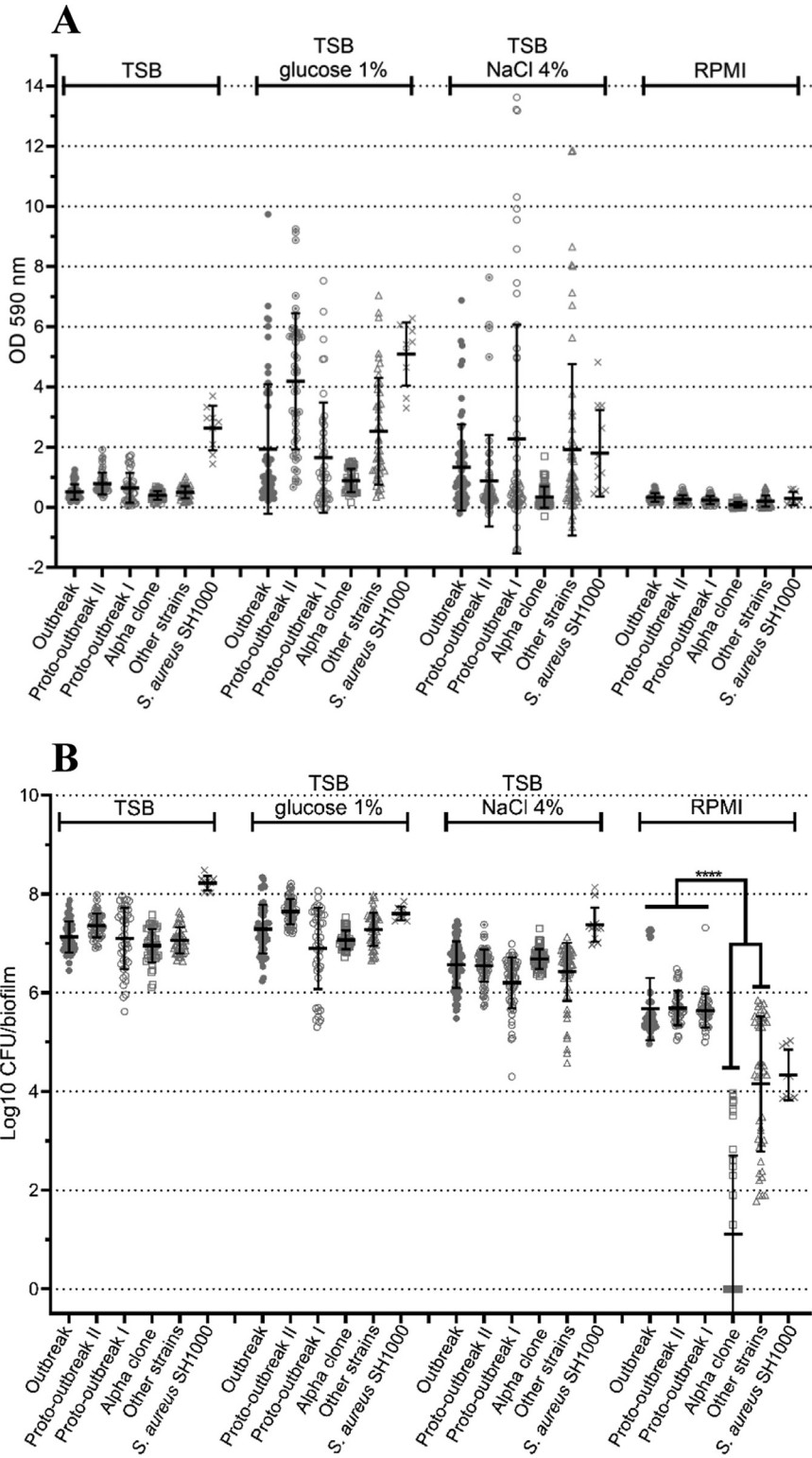

**FIG 2** Evaluation of biofilm formation among the five *S. capitis* groups under different growth conditions. (A) Total biomass (bacteria and biofilm matrix) evaluation by crystal violet staining. (B) Bacterial density in biofilms obtained by enumeration. The TSB medium allowed biofilm production under standard *in vitro* conditions, whereas TSB plus 1% glucose, TSB plus 4% NaCl, and RPMI medium represented a resource-rich environment, a hyperosmotic condition, and a nutritive stress condition, respectively. Results are shown as the mean $OD_{590}$ or number of CFU/biofilm for each group $\pm$ standard deviation. Statistical analyses were carried out using the Mann-Whitney nonparametric test with an $\alpha$ of 0.05.

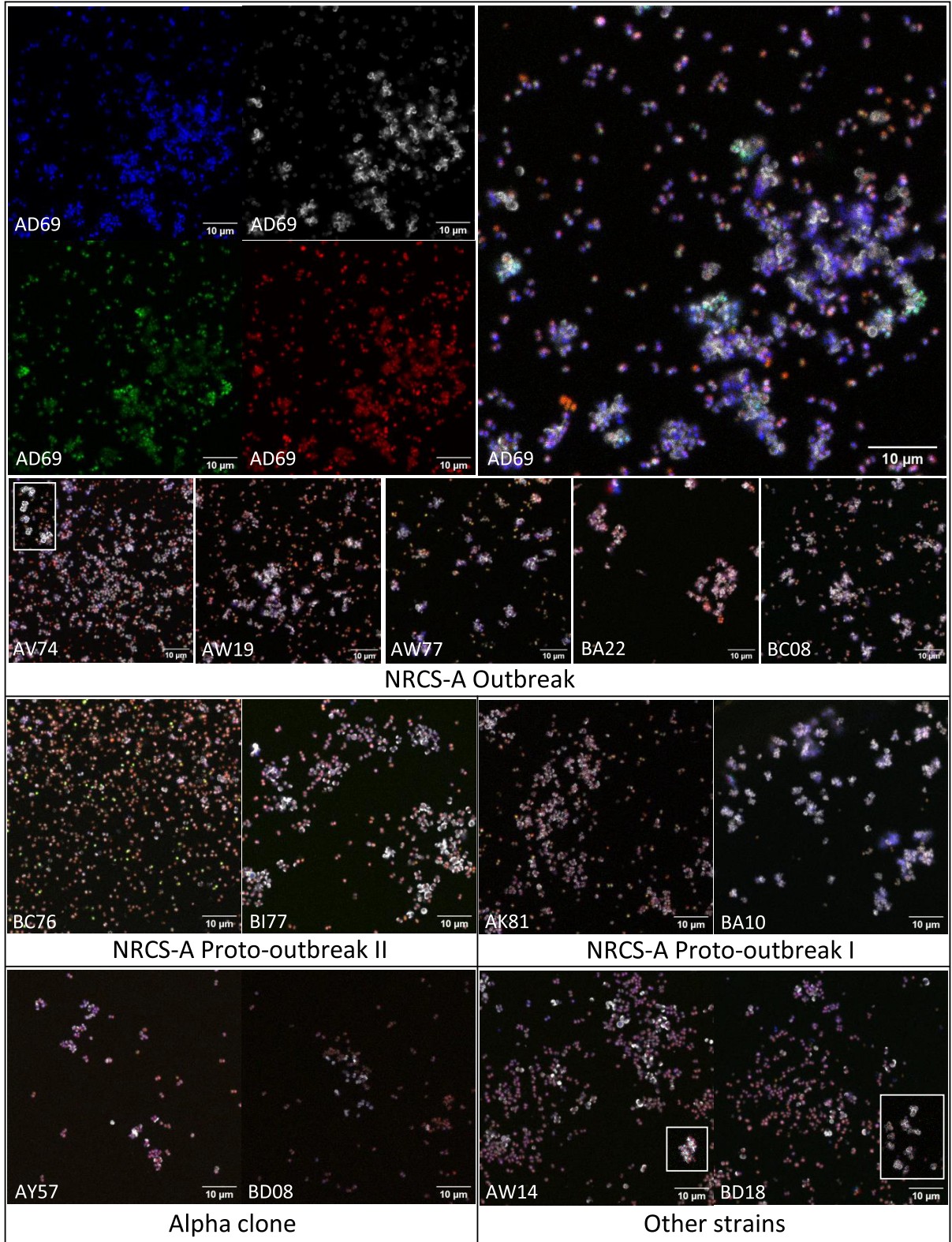

**FIG 3** *S. capitis* biofilm structure and matrix composition under nutrient stress. After biofilm formation under nutrient stress (RPMI medium), the different components of the matrix were marked and biofilms were observed using a Zeiss LSM 880 confocal microscope with a ×63 magnification. Blue, total DNA marked by Hoechst 33342; green, eDNA marked by TOTO-1 iodide; white, polysaccharides including PIA marked by WGA Alexa Fluor 647; red, proteins marked by Sypro ruby. Image acquisition was done individually for the four different matrix components (four different channels), and composite images were constructed using ImageJ software by superposition of the four channels. For strains AV74, AW14, and BD18, the images showing the general appearance of the biofilm did not allow for observation of the aggregates that were visible in other images. Since several images were taken for each strain, images of aggregates formed by these strains were added and framed in white.

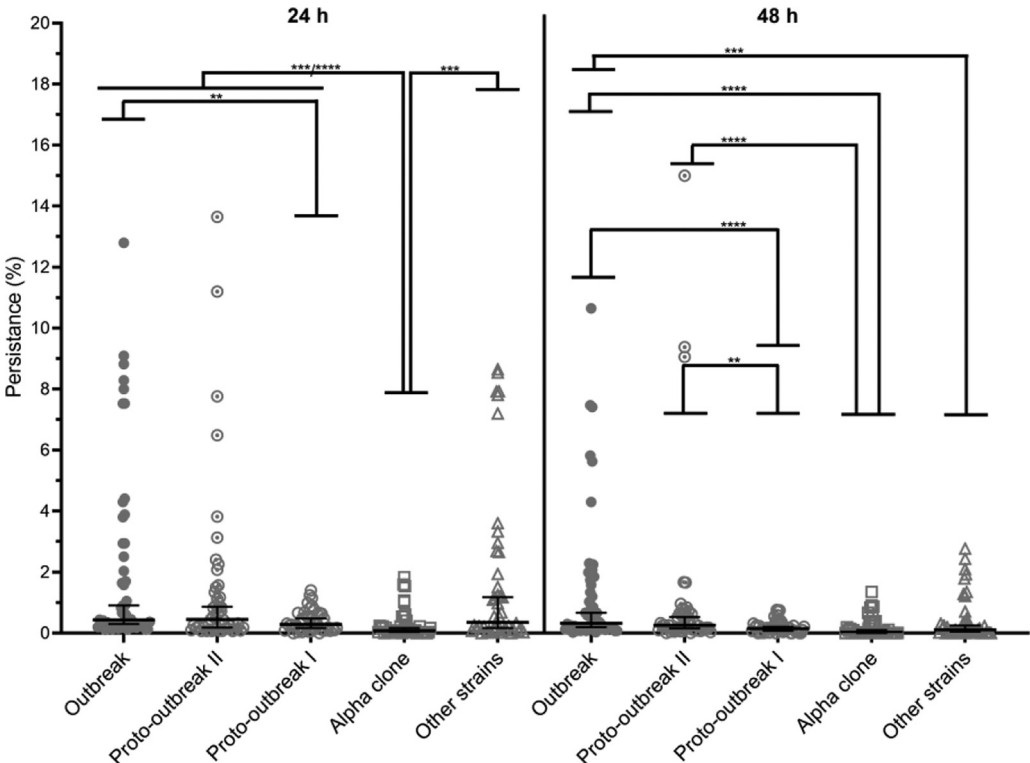

**FIG 4** Desiccation tolerance among the different groups of *S. capitis*. The percentage of persistence was obtained by comparison of the bacterial inocula at $T_0$ and after the desiccation stress ($T_{24}$ or $T_{48}$). Results are shown as the percentage of persistence of each group with 95% confidence intervals. Statistical analyses were carried out using the Mann-Whitney nonparametric test with an $\alpha$ of 0.05.

($P$ = 0.0001). The alpha clone showed also a significantly lower persistence (0.03%) than the outbreak (0.33%, $P$ < 0.0001) and proto-outbreak II (0.26%, $P$ < 0.0001) groups at 48 h. After this exposure time, it should also be noted that the proto-outbreak I group showed a significantly lower persistence (0.15%) than the outbreak and proto-outbreak II groups ($P$ < 0.0001 and $P$ = 0.0055, respectively) and that the "other strains" group showed a lower persistence (0.11%) than the outbreak group ($P$ = 0.0002). Complete eradication was never observed among the outbreak and proto-outbreak II groups.

**Effect of disinfectants.** The MIC and minimum bactericidal concentration (MBC) tests did not reveal different profiles of tolerance toward disinfectants (benzalkonium chloride [BAC], chlorhexidine digluconate [CDG], acriflavine [ACR], ethanol [Eth], and Surfanios Premium [SP]) between the different groups of *S. capitis* strains (Table 2). Of note, it was observed that the strains with at least one gene associated with disinfectant tolerance (*qacA*, *qacB*, *qacJ*, or *smr*) showed MICs and MBCs generally increased by one to two dilutions for BAC and ACR compared to strains without tolerance genes.

The exposure of bacterial strains to usual concentrations (concentrations typically used in a clinical context) of CDG, SP, and BAC reduced the inocula by more than 4 $\log_{10}$ (99.99%) after an exposure time of 5 min (Fig. 5). None of the tested disinfectants showed a better efficacy than that of others. However, tolerance to BAC after 60 min of contact was higher for the outbreak group (bacterial inoculum loss, 4.80 $\log_{10}$) than for the other groups ($P$ = 0.0273 for proto-outbreak II [bacterial inoculum loss, 5.51 $\log_{10}$], $P$ = 0.0005 for proto-outbreak I [bacterial inoculum loss, 6.33 $\log_{10}$], $P$ < 0.0001 for alpha clone [bacterial inoculum loss, 6.65 $\log_{10}$], and $P$ = 0.0356 for "other strains" [bacterial inoculum loss, 5.66 $\log_{10}$]).

## DISCUSSION

The pathogen *S. capitis* NRCS-A is well established in NICUs worldwide (5). Understanding the mechanisms involved in its ability to colonize and persist in these settings is crucial for the subsequent identification of effective means of control and eradication.

**TABLE 2** Disinfectants MICs and MBCs[a]

| Strain | MIC | | | | | MBC | | | | | Genes related to disinfectant tolerance | | | |
|---|---|---|---|---|---|---|---|---|---|---|---|---|---|---|
| | μg/mL | | | % | | μg/mL | | | % | | | | | |
| | BAC | CDG | ACR | Eth | SP | BAC | CDG | ACR | Eth | SP | qacA | qacB | qacJ | smr |
| ATCC 25923 BK12 | 2 | 1 | 32 | 6.25 | 0.002 | 4 | 2 | 32 | 12.5 | 0.002 | | | | |
| **NRCS-A Outbreak** | | | | | | | | | | | | | | |
| AD69 | 2 | 2 | 32 | 6.25 | 0.002 | 8 | 4 | 128 | 25 | 0.0039 | − | − | − | − |
| AV74 | 8 | 2 | 64 | 6.25 | 0.002 | 8 | 4 | 64 | 25 | 0.0039 | − | + | − | − |
| AW19 | 2 | 2 | 32 | 6.25 | 0.002 | 8 | 4 | 128 | 25 | 0.0039 | − | − | − | − |
| AW77 | 8 | 2 | 64 | 6.25 | 0.0039 | 16 | 4 | 128 | 25 | 0.0039 | + | − | − | − |
| BA22 | 4 | 1 | 8 | 12.5 | 0.002 | 4 | 2 | 32 | 25 | 0.0039 | − | − | − | − |
| BC08 | 2 | 2 | 32 | 6.25 | 0.002 | 8 | 2 | 64 | 25 | 0.002 | + | − | − | − |
| **NRCS-A proto-outbreak II** | | | | | | | | | | | | | | |
| AW20 | 4 | 2 | 32 | 6.25 | 0.002 | 8 | 2 | 128 | 25 | 0.002 | − | − | − | − |
| BC76 | 4 | 2 | 32 | 6.25 | 0.002 | 8 | 4 | 128 | 25 | 0.002 | − | − | − | − |
| BD01 | 2 | 2 | 32 | 6.25 | 0.002 | 8 | 2 | 128 | 25 | 0.002 | − | − | − | − |
| BD06 | 8 | 2 | 64 | 6.25 | 0.002 | 16 | 4 | 512 | 25 | 0.0039 | + | − | − | − |
| BI77 | 4 | 2 | 32 | 6.25 | 0.002 | 8 | 4 | 128 | 25 | 0.0039 | − | − | − | − |
| **NRCS-A proto-outbreak I** | | | | | | | | | | | | | | |
| AK81 | 4 | 2 | 32 | 6.25 | 0.002 | 8 | 2 | 64 | 25 | 0.0039 | − | + | − | − |
| BD62 | 8 | 2 | 32 | 6.25 | 0.002 | 16 | 2 | 64 | 25 | 0.0039 | + | − | + | + |
| BG77 | 1 | 1 | 16 | 6.25 | 0.002 | 8 | 2 | 32 | 25 | 0.002 | − | − | − | − |
| AL04 | 1 | 1 | 16 | 6.25 | 0.002 | 8 | 2 | 64 | 25 | 0.002 | − | − | − | − |
| BA10 | 0.5 | 0.5 | 8 | 12.5 | 0.001 | 8 | 2 | 64 | 25 | 0.0039 | + | − | − | + |
| **Alpha clone** | | | | | | | | | | | | | | |
| AY57 | 8 | 2 | 32 | 6.25 | 0.002 | 16 | 4 | 64 | 25 | 0.0039 | + | − | − | + |
| BD09 | 4 | 1 | 32 | 6.25 | 0.002 | 8 | 4 | 32 | 25 | 0.0039 | + | − | − | − |
| AX72 | 8 | 2 | 32 | 6.25 | 0.0039 | 16 | 2 | 64 | 25 | 0.0039 | + | − | − | − |
| AY24 | 8 | 2 | 64 | 6.25 | 0.0039 | 16 | 4 | 128 | 25 | 0.0039 | + | − | + | − |
| BD08 | 4 | 2 | 32 | 6.25 | 0.0039 | 8 | 2 | 64 | 25 | 0.0039 | + | − | − | − |
| **Other strains** | | | | | | | | | | | | | | |
| AW14 | 1 | 2 | 16 | 6.25 | 0.002 | 4 | 2 | 64 | 25 | 0.002 | − | − | − | − |
| BC70 | 4 | 2 | 32 | 6.25 | 0.002 | 4 | 4 | 128 | 25 | 0.0039 | − | − | + | − |
| BD28 | 8 | 1 | 32 | 6.25 | 0.002 | 16 | 2 | 64 | 25 | 0.002 | − | + | − | + |
| AY43 | 2 | 1 | 8 | 6.25 | 0.002 | 8 | 2 | 32 | 25 | 0.002 | − | − | − | + |
| BD18 | 8 | 2 | 64 | 6.25 | 0.0039 | 16 | 4 | 128 | 25 | 0.0039 | + | − | − | + |

[a]For each disinfectant tested, the intensity of the grey increases with increases in the MIC and MBC values. BAC, benzalkonium chloride; CDG, chlorhexidine digluconate; ACR, acriflavine; Eth, ethanol; SP, Surfanios Premium.

In this study, some of the mechanisms previously known to be involved in the environmental persistence of bacteria were studied and compared between strains belonging to the three subgroups of the NRCS-A clone and with other strains of *S. capitis*. No differences in biofilm production were shown in rich medium (TSB supplemented or not with glucose or NaCl) between strains of the NRCS-A clone and other *S. capitis* strains. Previous literature data about biofilms of *S. capitis* were discordant (7, 9, 19). Studies showing a better biofilm production in NRCS-A evaluated only the global biomass of biofilm by staining methods, whereas the present work also quantified the number of viable bacteria embedded in the biofilm matrix. In addition, the present study highlights the influence of the environment on biofilm development, since the biofilm formation of the different *S. capitis* groups was enhanced under glucose supplementation and a hyperosmotic condition and were decreased under nutrient stress. It is interesting to note that the different conditions of experimentation affected the global biomass production more and thus the matrix secretion than the number of bacteria. These results are consistent with previous studies which have shown that addition of glucose to TSB medium, and to a lesser extent the addition of NaCl, can enhance biofilm production in staphylococcal strains (20–22). In *S. capitis*, previous studies demonstrated that glucose supplementation and hyperosmotic stress enhanced biofilm production of clinical

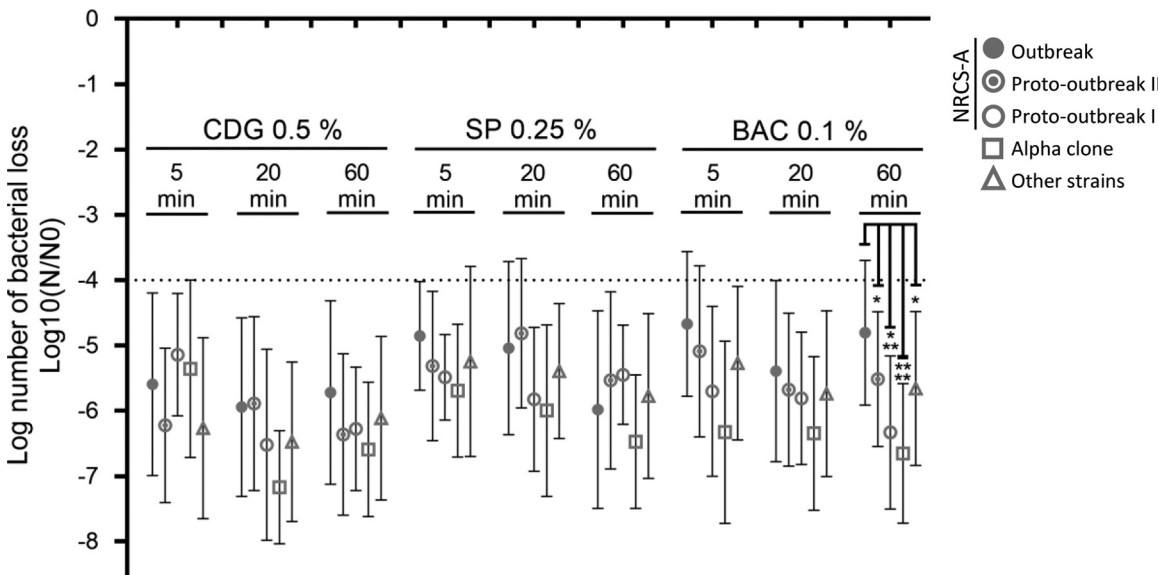

**FIG 5** Effect of disinfectants on *S. capitis* at usual concentrations and time of contact. Results are shown as the mean $\log_{10}(N/N_0)$ ± standard deviation, with $N$ equal to the remaining inoculum after treatment with disinfectant and $N_0$ equal to the remaining inoculum after treatment with water. Statistical analyses were carried out using the Mann-Whitney nonparametric test with an $\alpha$ of 0.05. CDG, chlorhexidine diglucanate; SP, Surfanios Premium; BAC, benzalkonium chloride.

isolates (23–25). Qu et al. found that the biofilm formation of *S. capitis* strains isolated from a NICU, including strains belonging to the NRCS-A clone, was not activated by glucose but was enhanced in response to hyperosmotic stress that activated the *icaADBC* operon and the development of a polysaccharidic matrix (25). The activation of the *ica* operon transcription in the presence of NaCl was also observed in *S. epidermidis* isolates from neonates (22). Interestingly, important variations in biofilm biomass with visual detachment after the washing step were observed in the present work when biofilm developed in NaCl-supplemented medium. This finding is consistent with the study of Lade et al. about *S. aureus* strains, in which the authors suggested that these variations could result from the loss of bacterial attachment to the well surface with excess NaCl (21).

To mimic a nutrient stress condition, we used RPMI medium because, despite containing different type of nutrients, this medium is specifically designed for eukaryotic cell culture and represents a nutrient-poor support for bacterial growth. Thus, we considered that the RPMI medium could mimic *in vitro* a NICU surface soiled during use by patients' secretions or by handling by caregivers and, in particular, incubators that are moistened and heated. Of note, in NICUs, incubators are disinfected only every 7 to 10 days but are cleaned daily using sterile water-soaked wipes (10). Under this nutrient stress condition, a decreased biofilm formation was observed for each *S. capitis* group. It is worth noting that the three subgroups of the NRCS-A clone showed a better tolerance to this stress with a higher number of cultivable bacteria. Similarly, the RPMI medium was shown by Wijesinghe et al. to allow increased adhesion of *Staphylococcus aureus* and *Pseudomonas aeruginosa* despite its low support for growth (26). As reported in the literature, one of the characteristics of the NRCS-A clone is the presence of a cluster of genes related to teichoic acid biosynthesis (*ispD*, *tarJ*, *tarK*, and *tagF*) (7, 8). Teichoic acids are involved in biofilm formation by taking part in the initial bacterial attachment (12). Regarding our results, it can be supposed that the presence of those genes in NRCS-A may positively affect its ability to adhere to surfaces under conditions of nutritive stress, favoring its persistence in the NICU environment. Additional experiments are needed to validate this hypothesis and to better understand the role of teichoic acids in NRCS-A adhesion to surfaces. Microscopic data revealed that under this nutrient stress, the different *S. capitis* strains formed dispersed small aggregates among scattered adherent bacteria, resulting in a very thin and immature biofilm. These observations were consistent with the total biomass evaluated, which was extremely low for strains of all groups. This could be explained either by the very immature

state of the biofilm or by the impossibility of developing a better one without a sufficient nutrient supply. The enhanced amount of visible polysaccharides among aggregates suggests the secretion of a PIA matrix, as already observed for *S. capitis* (25, 27). Some bacteria outside the aggregates also presented a polysaccharidic coverage that could be related to the labeling of the cell wall polysaccharides by the wheat germ agglutinin (WGA) Alexa Fluor 647 or by secretion of PIA by planktonic cells, as previously described for *S. aureus* (28). Of note, it would also be interesting to assess the biofilm matrix composition in a rich medium favoring its secretion. In this context, the method used to evaluate the matrix composition could be improved by using products specifically disrupting the different components of the matrix before the staining was carried out. This method allows the precise determination of the major components of the biofilm matrix by visualizing their targeted degradation by specific treatments (proteinase K and DNase I for proteins and eDNA and sodium metaperiodate for polysaccharides). Using this protocol, Panda and Singh precisely observed and described the composition of the biofilm matrix of clinical strains of *Staphylococcus haemolyticus* (29).

Other mechanisms of environmental persistence were explored in our study, but no specificities of the NRCS-A clone were observed in comparison to the alpha clone and the "other strains" groups. The desiccation stress assessment revealed a particular susceptibility of the alpha clone compared to that of the other four groups, whereas the NRCS-A outbreak and proto-outbreak II groups were never eradicated after 48 h. However, these results should be interpreted cautiously, since the statistical significance after 24 h of desiccation between the alpha clone and the NRCS-A proto-outbreak I subgroup seems not sufficient to be biologically relevant. Within the NRCS-A clone, the outbreak and proto-outbreak II groups were less impacted by the desiccation stress than was the proto-outbreak I group, despite heterogeneous results within each subgroup. This may reflect the evolution in environmental persistence within the NRCS-A clone leading to the successful establishment of the most recent subgroups in the NICUs (7). Previous studies reported that most Gram-positive bacteria such as *Staphylococcus*, *Streptococcus*, and *Enterococcus* can survive for months on dry surfaces in the case of a high level of relative humidity, low temperature, and high inoculum (30). On fabrics and plastics commonly used in hospitals, streptococci and staphylococci can survive for days to months after drying (31). On cotton, *S. aureus* was shown to better survive at room temperature with a low relative humidity (52%) than under a condition mimicking proximity with the human body (35°C, 83% relative humidity). One possible explanation is the decreased bacterial metabolism at low temperature that could induce a dormant state contributing to survival on dry surfaces (32). Some strains of coagulase-negative staphylococci were shown to persist for more than 90 days on polyethylene plastic, a material that can be encountered on neonatal incubators at room temperature and low relative humidity (30% to 49%) (31). On dry, previously disinfected hospital surfaces from an intensive care unit, Hu et al. identified the presence of persistent bacteria embedded in polymicrobial biofilms on more than 90% of surfaces (33).

Another hypothesis to explain persistence in NICUs despite disinfection is a decreased susceptibility to disinfectant molecules. Data about disinfectant tolerance are limited, given that there is neither consensus on the method to be used to assess the susceptibility of bacteria nor defined breakpoints for classifying a strain as susceptible or resistant (15, 34–36). However, in this study, no reduced susceptibility was observed for the NRCS-A clone compared to that of the other groups tested, and the MICs and MBCs never reached the concentration used in the clinical context. Conversely, Carter et al. observed an increased chlorhexidine MIC for NRCS-A strains compared to that for other *S. capitis* strains (9). Some studies have proposed a limit on concentrations to class staphylococci according to their MICs for disinfectants; reduced susceptibility was estimated at a MIC of ≥4 mg/L for BAC and chlorhexidine (16, 34–36) and a MIC of ≥16 mg/L for ACR (16, 37). According to these values, several strains of *S. capitis* with no group distinction showed reduced susceptibility to BAC and ACR in our study, in particular, the strains that contained in their genome genes associated with disinfectant tolerance. That is consistent with the results of Lepainteur et al., who detected reduced susceptibility to these disinfectants among coagulase-negative staphylococci isolated from catheter-associated bloodstream infections in very preterm neonates (16). To get closer to the conditions of hospital use, disinfectants

were tested at in-use concentrations and time of contact. All the disinfectants were extremely efficient from 5 min of contact. However, a large bacterial reduction is not sufficient to prevent bacterial regrowth, as shown on *S. aureus* biofilm (38). So, we can hypothesize that in NICUs, a few *S. capitis* strains are able to survive disinfection following adherence to dry surfaces and then to be transmitted from the environment to neonates. The present study was not designed to correlate the presence/absence of genes associated with disinfectant tolerance inside each *S. capitis* group with the disinfectant tolerance. This latter question is a complex matter, as previous studies have shown that genes associated with disinfectant tolerance, such as *qacA* and *qacB* encoding efflux pumps, do not always result in an increased bacterial tolerance to disinfectants like BAC and CDG nor do they systematically allow for cross-tolerance to different disinfectants (16, 39–41).

In this study, the alpha clone was the group that showed the lowest colonizing abilities with a higher doubling time and the lowest ability to form biofilm under nutrient stress and to persist on dry surfaces. The doubling time is generally even longer in the environment (42), which could make easier the elimination of a low inoculum of *S. capitis* belonging to the alpha clone during surface cleaning and disinfection. Thus, these characteristics could in part explain why this clone has not become endemic worldwide like the NRCS-A clone has.

Despite preventative measures to avoid pathogen cross-contamination, such as frequent surface decontaminations and cautious handwashing, the presence of pathogens and outbreaks involving different bacterial species including NRCS-A still occur in NICUs (10, 43–49). Innovative procedures of decontamination are needed to be more efficient than disinfectants in combatting bacterial persistence in NICUs and especially biofilm-embedded and adherent strains. In particular, it is crucial to develop new decontamination techniques that are both effective for bacterial eradication without a corrosive effect on surfaces and innocuous for neonates and health care workers. Moreover, it might be relevant to evaluate the impact of different materials on the ability of the clone NRCS-A to adhere and colonize. This approach could lead to revisions in the design of incubators and other medical equipment, in collaboration with manufacturers.

This study has limitations. First, the strains were selected to represent the general diversity among each group. This choice led to important intragroup variations despite the clonal nature of the NRCS-A clone involving difficulties comparing the different groups; the inclusion of more strains could have decreased this phenomenon. Second, this model of a single-species biofilm does not represent the complexity of environmental biofilm, which is mostly polymicrobial, and enumeration by culture methods does not allow detection of the viable but nonculturable bacteria present in this environment (33). Finally, desiccation stress was tested for 48 h only, and a longer period would be better for identifying a tolerance difference between groups. Moreover, it was not possible to monitor nor to strictly control experimental conditions of temperature and relative humidity.

**Conclusions.** Among the different mechanisms involved in bacterial colonization and persistence tested in this study, a better ability to adhere and form an extremely thin biofilm under conditions of nutritional stress and an ability to survive a 2-day desiccation have been identified in the NRCS-A strains. Future research about environmental decontamination against NRCS-A should take into account these two mechanisms.

## MATERIALS AND METHODS

**Strains and growth conditions.** Strains were obtained from the collection of the French National Reference Centre for Staphylococci (Lyon, France) and were conserved either in cryotubes (LaboModerne, Gennevilliers, France) at −20°C or as frozen inocula to a known concentration in TSB (bioMérieux, Marcy l'Etoile, France) with 10% glycerol (Avantor VWR International, Rosny-sous-Bois, France) at −80°C. From this collection, 26 strains of *S. capitis* isolated from blood cultures and previously sequenced were selected to represent the diversity in terms of genes associated with biofilm formation and with tolerance to disinfectant molecules among different clonal groups (Table 1) (7). The genomic sequences of these strains are available in the Sequence Read Archive under accession number PRJNA493527 (7). Five strains per group were thus selected from the outbreak, proto-outbreak II, and proto-outbreak I subgroups (belonging to the NRCS-A clone), from the alpha clone group (including *S. capitis* strains distinct from NRCS-A but also responsible for neonatal sepsis), and from a group named "other strains" that included nonclonal *S. capitis* strains not belonging to those two clones. We also included the CR01 strain, which is the reference strain for the subgroup *S. capitis* NRCS-A outbreak (50). The strain *S. aureus* SH1000 was used as a positive biofilm

producer control, and *S. aureus* ATCC 25923 was used as a control for the evaluation of MICs of disinfectant molecules (35).

**Growth kinetics.** From overnight cultures in TSB, bacterial suspensions were adjusted to an optical density at 600 nm ($OD_{600}$) of 0.05 in TSB. One hundred microliters of this suspension was then added in duplicate to a microplate (Greiner Bio-One, Cap Horn, France). The microplate was incubated for 24 h at 37°C in a Tecan Infinite 200 PRO microplate reader (Tecan, Männedorf, Swiss), with $OD_{600}$ reads every 15 min after an 8-s agitation. *S. aureus* strain SH1000 was used as a control with an already known doubling time (51). TSB medium without bacteria was used as a negative control, and its mean $OD_{600}$ value was subtracted at each time from that of the wells containing the staphylococcal strains. The experiments were performed four times in duplicate for each strain.

**Biofilm formation.** The abilities to form biofilm were compared between the five groups of *S. capitis* by previously published methods with some modifications (25, 52). Biofilms were grown in 96-well microplates in different culture media mimicking the following specific conditions: (i) a standard TSB medium containing 0.25% glucose (TSB), (ii) a standard TSB medium supplemented with additional glucose (TSB plus 1% glucose [Thermo Fisher Scientific, Waltham, MA, USA]), reflecting a high-resource environment, (iii) a hyperosmotic environment (TSB plus 4% NaCl [Thermo Fisher Scientific, Waltham, MA, USA]), and (iv) a nutritionally deprived environment (RPMI 1640 without phenol red [Thermo Fisher Scientific, Waltham, MA, USA] that, even if containing some nutrients, is specifically designed for eukaryotic cell culture and is considered a nutrient-poor medium for bacterial growth). For each condition, the wells of two different microplates were inoculated with $3 \times 10^5$ CFU from frozen inocula of known concentration diluted in the medium of interest. After 24 h at 37°C in a wet chamber, biofilms were washed to remove planktonic bacteria by using steam technology, a soft washing method that was designed to preserve the biofilm integrity, as described by Tasse et al. (52). One microplate was used for bacterial enumeration, and the other was used for total biomass evaluation (including both bacteria and biofilm matrix). For bacterial enumeration, biofilm-forming bacteria were recovered by scraping and suspended in 200 $\mu$L of 1× PBS (Thermo Fisher Scientific, Waltham, MA, USA). Bacteria were then enumerated by spotting 10 $\mu$L of serial dilutions on a Columbia agar plate with 5% sheep blood (COS) (bioMérieux, Marcy l'Etoile, France). For biomass evaluation, the biofilm was stained with 150 $\mu$L of crystal violet (ELITechGroup, Logan, UT, USA) for 10 min and then washed using the steam technology and dried. The stain was solubilized for 15 min in 33% acetic acid (Carlo Erba Reagents, Val-de-Reuil, France), and the $OD_{590}$ was read in a Tecan Infinite 200 PRO microplate reader. *S. aureus* strain SH1000 was used as a positive biofilm producer control, and medium alone was used as a negative control. The experiments were performed three times in triplicate.

**Biofilm matrix imaging under nutritional stress.** The six strains belonging to the outbreak subgroup of the NRCS-A clone and two strains of each of the other *S. capitis* groups were chosen to study the composition of their biofilm matrix in RPMI medium mimicking the environment of NICUs (Table 1). After 24 h of incubation as described above, four fluorescent markers were used to specifically stain the different components of the biofilm matrix: (i) TOTO-1 iodide (Thermo Fisher Scientific, Waltham, MA, USA), which fixes the extracellular DNA (eDNA) (53), (ii) Hoechst 33342 (Merk, Darmstadt, Germany), a cell-permeable stain that fixes the total DNA (54), (iii) wheat germ agglutinin (WGA) Alexa Fluor 647 (Thermo Fisher Scientific, Waltham, MA, USA), a lectin that fixes to *N*-acetyl-D-glucosamine and *N*-acetylneuraminic acid residues (55), and (iv) Sypro ruby (Thermo Fisher Scientific, Waltham, MA, USA), which fixes proteins (55). After washing by the steam method to remove planktonic bacteria, 100 $\mu$L of a mix of TOTO-1 iodide, Hoechst 33342, and Sypro ruby at concentrations of 1 $\mu$mol/L, 10 mg/L, and pure, respectively, was added to the biofilm and incubated for 30 min in the dark before washing by the steam method. Biofilms were then stained with 100 $\mu$L of WGA Alexa Fluor 647 at 5 mg/L for 15 min in the dark and then washed by the steam method. The biofilm was finally fixed with 100 $\mu$L of 4% paraformaldehyde (Thermo Fisher Scientific, Waltham, MA, USA) for 30 min in the dark, washed one time with distilled water using a pipette, and conserved in 50 $\mu$L of distilled water until microscopic observations. Biofilms were observed using a Zeiss LSM 880 confocal microscope (Zeiss, Oberkochen, Germany) coupled with ZEN (black edition) software (Zeiss, Oberkochen, Germany) with a 63-fold magnification. Analyses of the acquisitions were done using ImageJ software version 1.53q (National Institutes of Health, Bethesda, MD, USA). The experiments were performed two times.

**Impact of desiccation on *S. capitis* persistence.** Resistance to desiccation was explored as previously described, with some modifications (18). Briefly, after overnight growth in brain heart infusion broth (BHI) (bioMérieux, Marcy l'Etoile, France) at 37°C, a bacterial suspension in 1× PBS was adjusted to an $OD_{600}$ of 0.1 for each strain. To mimic the initial step of surface contamination, polypropylene microcentrifuge tube caps were inoculated with 20 $\mu$L of these suspensions (approximately $3 \times 10^5$ CFU). Inocula were then dried under a microbiological safety station for 3 h and then incubated for 24 to 48 h at room temperature in a nonhermetic closed box with protection from contamination but allowing air circulation. All the strains were tested at the same time and placed in the same nonhermetic box to ensure the same environmental conditions of temperature and humidity. The suspensions were enumerated at the time of inoculation ($T_0$). Then, to enumerate the persistent bacteria after the drying stress at 24 h ($T_{24}$) and 48 h ($T_{48}$), 1 mL of 1× PBS was added to each microcentrifuge tube, which was inverted and incubated for 15 min at room temperature. Microcentrifuge tubes were vortexed four times for 5 s each time, and then bacteria were diluted and enumerated by spotting on a COS agar plate. The percentage of persistence at each time was calculated from the initial inoculum count. The experiments were repeated three times in triplicate.

**Sensitivity to disinfectant molecules.** The MICs and MBCs of the *S. capitis* strains were determined for five different chemical disinfectants: BAC (Merk, Darmstadt, Germany), CDG (Thermo Fisher Scientific, Waltham, MA, USA), ACR (Merk, Darmstadt, Germany), Eth (Avantor VWR International, Rosny-sous-Bois, France), and SP [composed of *N*-(3-aminopropyl)-*N*-dodecylpropane-1,3-diamine and didecyldimethylammonium chloride] (ANIOS, Lezennes, France). MICs were determined using the microdilution method in accordance with the recommendations of the CLSI (Clinical and Laboratory Standards Institute) (https://clsi.org; retrieved 9 August

2022). For each disinfectant, a concentration range was tested by making serial 2-fold dilutions in Mueller-Hinton broth (MH) (Merk, Darmstadt, Germany) or in MH cation adjusted with 2% dimethyl sulfoxide (DMSO) (Merk, Darmstadt, Germany) for ACR in a 96-well microplate. A bacterial suspension in MH (or MH cation adjusted with 2% DMSO for ACR) from an overnight culture on a COS agar plate was inoculated into each well of this microplate to reach a bacterial concentration of approximately $5 \times 10^5$ CFU/mL. The concentrations of chemical disinfectant ranged from 128 $\mu$g/mL to 0.125 $\mu$g/mL for BAC and CDG, from 512 $\mu$g/mL to 0.5 $\mu$g/mL for ACR, from 50% to 0.049% for Eth, and from 0.25% to 0.0002% for SP. Absence of contamination was checked using control wells with MH alone and diluted disinfectant alone. Bacterial growth was checked in control wells without disinfectants. *S. aureus* ATCC 25923 was used as a control with known MICs and MBCs for BAC, CDG, and ACR (35). The plates were incubated for 18 h at 37°C in a wet chamber. The MIC was determined as the lowest concentration of chemical disinfectant preventing visible bacterial growth. To determine the MBC, 10-$\mu$L suspensions in wells with disinfectant concentrations greater than or equal to the MIC were spotted on COS agar plates and then incubated overnight at 37°C. The MBC was determined as the lowest concentration of chemical disinfectant resulting in a decrease of at least 3 $\log_{10}$ (99.9%) of the initial inoculum (56).

**Effect of usual concentrations of disinfectants.** The effect of chemical disinfectants on each strain of each of the five *S. capitis* groups was evaluated by mimicking their conditions of use in a clinical context. The protocol was adapted from a report by Tremblay et al. (57). From an overnight culture in BHI, bacteria were diluted at a 2 McFarland standard; from this dilution, 250 $\mu$L was added to a 2-mL microcentrifuge tube. The tubes were centrifuged, and the supernatant was discarded. For disinfectant treatment, the remaining pellets were suspended in 50 $\mu$L of either (i) water or (ii) disinfectant diluted in water at concentrations used in a clinical context (CDG at 0.5% [$5 \times 10^3$ $\mu$g/mL] [58], SP at 0.25%, or BAC at 0.1% [$1 \times 10^3$ $\mu$g/mL] [59]) for 5 min, 20 min, or 60 min at room temperature. To control the treatment durations, the action of the disinfectants was stopped by the addition of 1.8 mL of Dey Engley neutralizing broth (Merk, Darmstadt, Germany), followed by three successive washings with Dey Engley neutralizing broth. The washed pellets were then suspended in 250 $\mu$L of BHI, and the bacteria were diluted for subsequent enumeration by the spot method. The results were expressed as log number of bacterial loss and were determined as follows: $\log_{10}(N/N_0)$, with $N$ equal to the bacteria remaining after treatment with the disinfectant and $N_0$ equal to the remaining bacteria after treatment with water. The experiments were repeated three times in triplicate. The efficacy of disinfectant neutralization was assessed by treating the bacterial pellet with 50 $\mu$L of neutralized disinfectant under the same conditions.

**Graphic representation and statistical analyses.** The graphics were drawn using GraphPad Prism 8 (GraphPad, La Jolla, CA, USA). Means with standard deviations were used to present data for biofilm formation and for the effect of the different disinfectants at the usual concentration and are represented, respectively, as scatter dot plots and symbols representing the mean. According to the dispersal of the results, we chose to represent the results of growth kinetics and effect of desiccation using the median with 95% confidence interval. Statistical analyses were performed using GraphPad Prism 8 and the Mann-Whitney nonparametric test with an $\alpha$ risk of 0.05.

**Data availability.** Data are available from the corresponding author on request.

## ACKNOWLEDGMENTS

We thank the ANR (Agence Nationale de la Recherche), which supported this work as part of the project NeoSCap ANR 19-CE17-0004-01, and also the microscopy platform CIQLE (Centre d'Imagerie Quantitative Lyon-Est) for their help and support.

The funders had no role in study design, data collection and interpretation, or the decision to submit the work for publication.

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
