## [Reviewer comments · Microbiology Spectrum]

Microbiology Spectrum

Environmental persistence of *Staphylococcus capitis* NRCS-A in neonatal intensive care units: role of biofilm formation, desiccation and disinfectants tolerance

Marie Chavignon, Ludivine Coignet, Mélanie Bonhomme, Marine Bergot, Anne Tristan, Paul Verhoeven, Jérôme Josse, Frédéric Laurent, and Marine Butin

Corresponding Author(s): Marine Butin, Neonatal Intensive Care unit, Hospices civil de Lyon

Review Timeline:

Submission Date:

October 20, 2022

Accepted:

November 2, 2022

Editor: Cezar Khursigara

Reviewer(s): The reviewers have opted to remain anonymous.

Transaction Report:

DOI: <https://doi.org/10.1128/spectrum.04215-22>

November 2, 2022

Dr. Marine Butin
Neonatal Intensive Care unit, Hospices civil de Lyon
Lyon
France

Re: Spectrum04215-22 (Environmental persistence of *Staphylococcus capitis* NRCS-A in neonatal intensive care units: role of biofilm formation, desiccation and disinfectants tolerance)

Dear Dr. Marine Butin:

Your manuscript has been accepted, and I am forwarding it to the ASM Journals Department for publication. You will be notified when your proofs are ready to be viewed.

Sincerely,

Cezar Khursigara
Editor, Microbiology Spectrum
